# Description and Whole-Genome Sequencing of *Mariniflexile litorale* sp. nov., Isolated from the Shallow Sediments of the Sea of Japan

**DOI:** 10.3390/microorganisms12071413

**Published:** 2024-07-12

**Authors:** Lyudmila Romanenko, Evgeniya Bystritskaya, Yuliya Savicheva, Viacheslav Eremeev, Nadezhda Otstavnykh, Valeriya Kurilenko, Peter Velansky, Marina Isaeva

**Affiliations:** 1G.B. Elyakov Pacific Institute of Bioorganic Chemistry, Far Eastern Branch, Russian Academy of Sciences, Prospect 100 Let Vladivostoku, 159, Vladivostok 690022, Russia; ep.bystritskaya@yandex.ru (E.B.); iu.savicheva0@yandex.ru (Y.S.); wieremeew@gmail.com (V.E.); chernysheva.nadezhda@gmail.com (N.O.); valerie@piboc.dvo.ru (V.K.); 2A.V. Zhirmunsky National Scientific Center of Marine Biology, Far Eastern Branch, Russian Academy of Sciences, Palchevskogo Street 17, Vladivostok 690041, Russia; velansky.pv@gmail.com

**Keywords:** *Mariniflexile litorale* sp. nov., *Bacteroidota*, marine bacteria, shallow sediments, chromosome, pan-genome analysis

## Abstract

A Gram-negative, aerobic, rod-shaped, non-motile, yellow-pigmented bacterium, KMM 9835^T^, was isolated from the sediment sample obtained from the Amur Bay of the Sea of Japan seashore, Russia. Phylogenetic analyses based on the 16S rRNA gene and whole genome sequences positioned the novel strain KMM 9835^T^ in the genus *Mariniflexile* as a separate line sharing the highest 16S rRNA gene sequence similarities of 96.6% and 96.2% with *Mariniflexile soesokkakense* RSSK-9^T^ and *Mariniflexile fucanivorans* SW5^T^, respectively, and similarity values of <96% to other recognized *Mariniflexile* species. The average nucleotide identity and digital DNA–DNA hybridization values between strain KMM 9835^T^ and *M. soesokkakense* KCTC 32427^T^, *Mariniflexile gromovii* KCTC 12570^T^, *M. fucanivorans* DSM 18792^T^, and *M. maritimum* M5A1M^T^ were 83.0%, 82.5%, 83.4%, and 78.3% and 30.7%, 29.6%, 29.5%, and 24.4%, respectively. The genomic DNA GC content of strain KMM 9835^T^ was 32.5 mol%. The dominant menaquinone was MK-6, and the major fatty acids were iso-C15:0, iso-C15:1ω10c, and C15:0. The polar lipids of strain KMM 9835^T^ consisted of phosphatidylethanolamine, two unidentified aminolipids, an unidentified phospholipid, and six unidentified lipids. A pan-genome analysis showed that the KMM 9835^T^ genome encoded 753 singletons. The annotated singletons were more often related to transport protein systems (SusC), transcriptional regulators (AraC, LytTR, LacI), and enzymes (glycosylases). The KMM 9835^T^ genome was highly enriched in CAZyme-encoding genes, the proportion of which reached 7.3%. Moreover, the KMM 9835^T^ genome was characterized by a high abundance of CAZyme gene families (GH43, GH28, PL1, PL10, CE8, and CE12), indicating its potential to catabolize pectin. This may represent part of an adaptation strategy facilitating microbial consumption of plant polymeric substrates in aquatic environments near shorelines and freshwater sources. Based on the combination of phylogenetic and phenotypic characterization, the marine sediment strain KMM 9835^T^ (=KCTC 92792^T^) represents a novel species of the genus *Mariniflexile*, for which the name *Mariniflexile litorale* sp. nov. is proposed.

## 1. Introduction

The genus *Mariniflexile*, with the type species *Mariniflexile gromovii*, was proposed by Nedashkovskaya et al. [1] and subsequently emended by Jung et al. [2], Jung, Yoon [3], and Park et al. [4]. The genus *Mariniflexile* belongs to the family *Flavobacteriaceae* [5], phylum *Bacteroidota* [6], and currently contains seven species with validly published names as listed at https://lpsn.dsmz.de/genus/mariniflexile (accessed on 3 May 2024) [7]. Bacteria of the genus *Mariniflexile* have been isolated from diverse marine sources, including sea urchin *Strongylocentrotus intermedius* [1], seawater [2,8], seawater at its confluence with a freshwater source [3,4], and an oyster [9]. It has been reported that *Mariniflexile fucanivorans*, a species recovered from mud formed in the process of recycling the effluent of an alginate-extraction plant, was able to degrade sulfated fucans of brown algae [10]. A unique feature of *Bacteroidota* is the presence of the type IX secretion system (T9SS), responsible for protein secretion through the outer membrane and involved in gliding motility, S-layer biogenesis, and biopolymer degradation [11]. In addition, the second adaptive system utilized uniquely by *Bacteroidota* is polysaccharide utilization loci (PULs), which allow bacteria to degrade polysaccharides in different environments. PULs are typically organized into contiguous loci where SusC/D transporter genes are adjacent to genes encoding transcriptional regulators and carbohydrate-active enzymes (CAZymes) [12].

In the present study, a Gram-negative, aerobic, non-motile bacterium, KMM 9835^T^, was isolated from the shallow sediment sample obtained from the Amur Bay of the Sea of Japan, Russia, and characterized. To obtain insights into the phylogenetic relationships and metabolic potential of the strain KMM 9835^T^ among *Mariniflexile* species, phylogenomic and pan-genomic analyses were performed. In addition, the genomic sequence of *Mariniflexile soesokkakense* KCTC 32427^T^ as a closely related phylogenetic neighbor was determined. Based on combined phylogenomic analyses and phenotypic properties, a novel species, *Mariniflexile litorale* sp. nov., is described.

## 2. Materials and Methods

### 2.1. Bacterial Strains

Strain KMM 9835^T^ was isolated from the shallow sediments sampled from the Amur Bay of the Sea of Japan, Russia (42°59′23.4″ N 131°29′07.8″ E) in July 2010 by a standard dilution plating method and incubated on marine agar 2216 (MA; BD Difco^TM^, Sparks, MD, USA) at 28 °C. This strain was grown aerobically on MA 2216 or in marine broth (MB) 2216 (BD Difco^TM^, Sparks, MD, USA) at 28 °C and stored at −70 °C in MB 2216 supplemented with 20% (*v*/*v*) glycerol. The strain KMM 9835^T^ has been deposited in the Collection of Marine Microorganisms (KMM), G.B. Elyakov Pacific Institute of Bioorganic Chemistry, Far Eastern Branch, Russian Academy of Sciences, Vladivostok, Russia, and in the Korean Collection for Type Cultures (KCTC), Korea, as KCTC 92792^T^. The type strains *M. soesokkakense* KCTC 32427^T^ and *Mariniflexile maritimum* KCTC 72895^T^ were purchased from the Korean Collection for Type Cultures, Korea, to be used in the comparative phenotypic tests.

### 2.2. Phenotypic Characterization

Oxidase and catalase reactions, Gram staining, and motility (the hanging drop method) were assessed following the previously described method [13]. Gliding motility was examined as described by Bowman [14]. The morphology of cells negatively stained with 1% phosphotungstic acid was observed on carbon-coated 200-mesh copper grids using the electronic transmission microscope Libra 120 FE (Carl Zeiss, Oberkochen, Germany), provided by the A.V. Zhirmunsky National Scientific Center of Marine Biology, Far Eastern Branch, Russian Academy of Sciences. The following physiological tests, including hydrolysis of gelatin, starch, casein, Tweens 20, 40, 80, DNA, chitin, L-tyrosine, and growth at different salinities (0–12% NaCl), temperatures (4–40 °C), and pH values (4.0–11.5), were studied on the artificial seawater (ASW)-based media as described earlier [15,16]. Biochemical characteristics and enzyme activities were measured by the API 20E, API 20NE, and API ZYM test strips (bioMérieux, Marcy-l’Étoile, France) according to the manufacturer’s instructions.

### 2.3. Chemotaxonomic Analyses

For the lipid analyses of bacterial strain KMM 9835^T^, *M. soesokkakense* KCTC 32427^T^ and *M. maritimum* KCTC 72895^T^ were cultivated on MA 2216 at 28 °C. Lipids were extracted according to the method of Folch et al. [17]. Two-dimensional thin layer chromatography of polar lipids was performed on Silica gel 60 F254 (10 × 10 cm, Merck, Darmstadt, Germany), applying chloroform–methanol–water (65:25:4, *v*/*v*) for the first direction, chloroform–methanol–acetic acid–water (80:12:15:4, *v*/*v*) for the second one [18], and spraying with specific reagents [19]. Fatty acid methyl esters (FAMEs) were prepared following the procedure of the Microbial Identification System (MIDI) [20]. A chromatograph (Shimadzu, Kyoto, Japan) with a flame ionization detector equipped with a SPB-5 capillary column (30 m × 0.25 mm × 0.25 mkm) was used. Identification of FAMEs was carried out by comparing the equivalent chain length values and retention times of the samples to those of the standards (Standard bacterial acid methyl ester mix 47080-U, Supelco, Bellefonte, PA, USA). In addition, FAMEs were investigated using a GC-MS Shimadzu QP2020 (Shimadzu, Kyoto, Japan) with a column Shimadzu SH–Rtx–5MS (30 m × 0.25 mm × 0.25 mkm) and a temperature program from 160 °C to 320 °C at a rate of 2 °C/min). To determine the position of double bonds and methyl groups, fatty acids were analyzed as 4,4-dimethyloxazoline derivatives [21] using GC–MS with the SH–Rtx–5MS column at a temperature from 180 °C to 320 °C, 2 °C/min. Menaquinones were examined by HPLC, as described by Hirashi et al. (1996) [22]. A Shimadzu LC–30 chromatograph with a photodiode array detector (SPD–M30A) was used. Absorption spectra of lipid extracts redissolved in methanol at a concentration of 0.2 mg/mL were obtained on a Shimadzu UV–2600 spectrophotometer. The presence of flexirubin pigments was determined as described by Fautz and Reichenbach [23].

### 2.4. 16S rRNA Gene Sequence and Phylogenetic Analysis

Genomic DNA of strain KMM 9835^T^ was extracted using the NucleoSpin Tissue kit (Macherey–Nagel, Düren, Germany) following the manufacturer’s instructions. The 16S rRNA gene was PCR-amplified using 27F (5′-AGAGTTTGATCMTGGCTCAG-3′) and 1492R (5′-TACGGTTACCTTGTTACGACTT-3′) primers with 5 min of denaturation (96 °C) followed by 25 cycles of 30 s of denaturation (95 °C), 30 s of annealing (55 °C), and 1 min 20 s of elongation (72 °C), finalizing with 5 min of elongation (72 °C). The obtained amplicons of standard bacterial acid were sequenced and compared with those of their closest relatives using the EzBioCloud service [24]. Phylogenies were performed on the GGDC web server (http://ggdc.dsmz.de/, accessed on 16 May 2024) [25] using the DSMZ pipeline [26] applied to a single gene. Maximum likelihood (ML) and maximum parsimony (MP) trees were inferred from the alignment with RAxML [27] and TNT [28], respectively. The neighbor-joining (NJ) tree was reconstructed with MEGA version 11 [29] using the Kimura two-parameter model of nucleotide substitutions. The robustness of phylogenetic trees was estimated by the bootstrap analysis of 1000 replicates.

### 2.5. Whole-Genome Sequencing, Phylogenomic, and Comparative Analyses

Genomic DNAs were extracted from strains KMM 9835^T^ and *M. soesokkakense* KCTC 32427^T^ by the NucleoSpin Tissue kit (Macherey–Nagel, Düren, Germany). The DNA quality was estimated by agarose gel electrophoresis, and the DNA quantity was measured on the Qubit 4.0 Fluorometer (Thermo Fisher Scientific, Singapore). The DNA libraries were prepared with Nextera DNA Flex kits (Illumina, San Diego, CA, USA) and were sequenced on an Illumina MiSeq instrument using paired-end runs with a 250-bp read length. The nanopore library was prepared for KMM 9835^T^ using EXP-NBD104 and SQK-LSK109 kits (Oxford Nanopore Technologies, Oxford, UK), according to the Native barcoding genomic DNA protocol. The reads were trimmed using Trimmomatic version 0.39 [30] and their quality assessed using FastQC version 0.11.8 (https://www.bioinformatics.babraham.ac.uk/projects/fastqc/, accessed on 21 August 2021 and 30 November 2023). Filtered reads of the *M. soesokkakense* RSSK-9^T^ genome were assembled into contigs with SPAdes version 3.15.3 [31], and genome metrics were calculated with QUAST version 5.0.2 [32]. Dorado version 0.4.3 (Oxford Nanopore Technologies, Oxford, UK) was used with default parameters to quality filter Nanopore reads and filter out sequences 1000 bp in length. Hybrid assembly of strain KMM 9835^T^ was performed using Unicycler v0.4.8 [33] with default parameters. Sequencing depth was estimated using samtools version 1.3 [34]. The genome completeness and contamination of strains KMM 9835^T^ and RSSK-9^T^ were estimated by CheckM version 1.1.3 based on the taxonomic-specific workflow (lineage *Flavobacteriales*) [35].

Comparisons of the Average Nucleotide Identity (ANI), Average Amino Acid Identity (AAI), and digital DNA–DNA hybridization (dDDH) values of the strains KMM 9835^T^ and KCTC 32427^T^ with their closest neighbors were performed with the online servers ANI/AAI–Matrix [36] and TYGS platform [37], respectively. The phylogenomic analysis was performed using PhyloPhlAn software version 3.0.1 based on a set of 400 conserved bacterial protein sequences using the RAxML program under the PROTCATLG model with bootstrapping of 100 replicates (flag −b +−100) [38].

Genome annotation was carried out using the NCBI Prokaryotic Genome Annotation Pipeline (PGAP) [39], Rapid Annotation using Subsystem Technology (RAST) [40], and Prokka [41]. The circular genome of KMM 9835^T^ was visualized using the Proksee platform [42]. Putative Horizontal Gene Transfer (HGT) events were detected via Alien Hunter [43]. CRISPR arrays and associated Cas proteins were found using CRISPR/Cas Finder [44]. Replication origin and terminus were predicted by Ori-Finder 2022 [45].

Identification of the Type 9 Secretion System (T9SS) components was conducted with MacSyFinder (TXSScan T9SS model) [46,47]. Conserved C-terminal domains (CTDs) of T9SS were searched using HMMER 3.4 (http://hmmer.org, E value of 1 × 10^−2^), and the hmm files of Type A (TIGR04183), Type B (TIGR04131), and ChiA (NF033708) were obtained from NCBI (accessed on 11 April 2024). CAZymes were predicted using the dbCAN3 meta server with default settings (https://bcb.unl.edu/dbCAN2/index.php, accessed on 9 December 2023) [48]. Genes predicted by no less than two algorithms integrated into the server were defined as CAZymes and selected for further analysis. CAZyme-containing gene clusters (CGCs) and PULs were annotated via the dbCAN-PUL meta server [49]. The relative abundances of CAZymes were visualized by heat maps using the pheatmap version 1.0.12 package in RStudio version 2022.02.0 + 443 with R version 4.1.3.

Metabolism estimation and pan-genomic analysis were conducted using anvi’o version 8 [50]. The FASTA files reformatted into contigs-fasta using the ‘anvi-script-reformat-fasta’ command were imported into anvi’o as contigs-db with the ‘anvi-gen-contigs-database’ command. These contigs-db’s were annotated via the ‘anvi-run-kegg-kofams’ command using the snapshot of the KEGG database accessed on 22 September 2023 [51]. An ‘anvi-estimate-metabolism’ command was consequently run with the ‘--include-metadata’ and ‘--matrix-format’ flags. The obtained data were analyzed manually using Microsoft Excel and compared with KAAS annotation data [52]. The pan-genome was reconstructed using the anvi’o workflow described at https://merenlab.org/2016/11/08/pangenomics-v2/ accessed on 28 March 2024. Cumulative curves were drawn using PanGP version 1.0.1 [53]. Pan-genome openness was estimated under Heap’s law model [54]. Fonts and sizes in all figures were edited manually in Adobe Photoshop CC 2018 for better visualization.

## 3. Results

### 3.1. Phylogenetic and Phylogenomic Analysis

Based on 16S rRNA gene sequence similarities calculated using the EzBioCloud service [24], strain KMM 9835^T^ (OQ300347, 1398 bp) was close to *M. soesokkakense* RSSK-9^T^ (96.6%), *Mariniflexile* sp. TRM1-10 (96.6%), *M. fucanivorans* SW5^T^ (96.2%), *Gaetbulibacter lutimaris* D1-y4^T^ (96.0%), and *Siansivirga zeaxanthinifaciens* CC-SAMT-1^T^ (96.0%). The other representatives of *Flavobacteriaceae* shared values less than 96.0%, including *Mariniflexile* type strains: *M. aquimaris* HWR-17^T^ (95.7%), *M. gromovii* KMM 6038^T^ (95.7%), *M. jejuense* SKK2-3^T^ (95.6%), and *M. ostreae* TYO-10^T^ (94.9%). *M. maritimum* M5A1M^T^ was absent from the list of the first 50 hits from the EzBioCloud 16S database.

On the 16S rRNA phylogenetic trees, the position of strain KMM 9835^T^ was uncertain due to low bootstrap support when *M. maritimum* M5A1M^T^ was included in the analysis (Figure 1). M5A1M^T^ exclusion and TRM1-10 addition resulted in KMM 9835^T^ clustering with *Mariniflexile* spp. strains (except for TYO-10^T^) under strong bootstrap support (Appendix A).

There are currently seven species of the genus *Mariniflexile* with validly published names, but only three genomes of type strains *M. gromovii*, *M. fucanivorans*, and *M. maritimum* are available. In this study, the genome sequence of the fourth type strain, *M. soesokkakense* KCTC 32427^T^, was obtained. The genomes of three type strains (DSM 18792^T^, KCTC 12570^T^, and M5A1M^T^) were retrieved from NCBI (Table 1). The phylogenomic tree based on concatenated sequences extracted from the genomes of *Mariniflexile* species and related taxa showed that strain KMM 9835^T^ formed a distinct line within the genus *Mariniflexile* (Figure 2).

The ANI/AAI values between the genomes of strain 9835^T^ and the type strains of *M. soesokkakense* KCTC 32427^T^, *M. gromovii* KCTC 12570^T^, *M. fucanivorans* DSM 18792^T^, and *M. maritimum* M5A1M^T^ were 83.0%/83.5%, 82.5%/81.7%, 83.4%/80.7%, and 78.3%/77.6%, respectively, which were lower than the 95−96% threshold value accepted for species delineation [55]. The dDDH values (formula d4) between strain KMM 9957^T^ and the four relatives, ranging from 24.4% (*M. maritimum* M5A1M^T^) to 30.7% (*M. soesokkakense* KCTC 32427^T^), were below the 70% threshold value accepted for species delineation [56,57]. These overall genomic relatedness indices (OGRIs) and phylogenomic position suggest that KMM 9835^T^ represents a novel species in the genus *Mariniflexile*.

### 3.2. Genomic Characteristics and Pan-Genome Analysis of the Mariniflexile Genus

The complete genome of strain KMM 9835^T^ was *de novo* assembled into one chromosome with an estimated size of 4,521,428 bp and an overall G+C content of 62.1%. The two genome-extracted 16S rRNA gene sequences were 100% identical to the PCR-amplified one (OQ300347). The genome contains 3752 protein coding sequences, 36 tRNAs, and 6 rRNA genes (two 16S-23S-5S operons). The indices recommended to evaluate the quality of the genomic data [58,59] are shown in Table 1. The observed characteristics satisfy the proposed minimal standards for the taxonomy of prokaryotes and indicate high genome quality. In total, seven *Mariniflexile* strains were taken for comparative genome analysis, two of which have been sequenced in this study (KMM 9835^T^ and KCTC 32427^T^). In addition to the genomes of three type strains (DSM 18792^T^, KCTC 12570^T^, and M5A1M^T^), two high-quality genomes of *Mariniflexile* sp. strains (AS56 and TRM1-10) were retrieved from NCBI. Their basic genome indices are listed in Table 1. The ML phylogenomic tree, including two last genomes, clearly showed that strains AS56 and TRM1-10 might present two novel species of the genus *Mariniflexile* (Appendix A). The genome sequences contain from 3157 (*M. maritimum* M5A1M^T^) to 4104 (*M. fucanivorans* DSM 18792^T^) genes, from 36 (KMM 9835^T^ and *M. soesokkakense* KCTC 32427^T^) to 41 (TRM1-10) tRNAs, and from one up to two *rrn* operon copies (KMM 9835^T^ and TRM1-10).

The KMM 9835^T^ chromosome map was built and visualized using Proksee [42] (Figure 3). Genome annotations were carried out using the RAST tool kit [40] and Prokka [41]. The first gene (*dnaA*) in the genome sequence was automatically assigned as the origin of replication; however, its position did not align with a GC skew plot. To identify the origin (*oriC*) and terminus (*ter*) of replication, the Ori-Finder 2022 server [45] was utilized. Regions adjacent to the *mnmG* and *leuB* genes were predicted as *oriC* and *ter*, respectively. These regions aligned with the GC skew plot and had the highest Ori-Finder scores. The chromosomal level of genomic assemblies obtained for strains KMM 9835^T^ and TRM1-10 made it possible to estimate the exact numbers of *rrn* operons (Table 1); both *rrn* operons of KMM 9835^T^ are located on the leading strands (Figure 3). The 13 retron-type RNA-directed DNA polymerase (EC 2.7.7.49) genes and one CRISPR-Cas region, among them seven full-length, were found in the genome strain KMM 9835^T^ (Figure 3). Three loci had truncated ORFs.

To determine genus-related features, a pan-genome analysis of *Mariniflexile* species (Table 1) was performed using orthologous clustering and metabolic pathway reconstruction with the anvi`o platform [50]. The *Mariniflexile* pan-genome (Figure 4) comprised a total of 9163 gene clusters (distance: Euclidean; linkage: Ward) with 25,933 gene calls. The core genome included 2202 core gene clusters covering 15,113 genes, of which 1625 were single-copy genes (SCGs). The accessory shell and cloud clusters were composed of 913 (3317 genes) and 1074 (2320 genes) clusters, respectively. A unique part of the pan-genome included 4974 gene clusters (5183 genes) of singletons. The largest and smallest numbers of singletons were observed in the genomes of *Mariniflexile* sp. TRM1-10 (1083 clusters) and *M. soesokkakense* KCTC 32427^T^ (377 clusters). The annotated singletons in the KMM 9835^T^ genome were more prevalent and often related to transport protein systems (SusC), transcriptional regulators (AraC, LytTR, and LacI), and enzymes (Glycosylases). According to the genome size modeling, the *Mariniflexile* pan-genome is open with a γ value of 0.58 (Appendix A).

The completeness of metabolic pathways for all *Mariniflexile* genomes was calculated using the anvi’o platform [50,51] and analyzed manually using Microsoft Excel and KAAS annotation data [52]. From these results, the pathways for glycan (lipopolysaccharide), lipid, and nucleotide metabolisms were complete in all *Mariniflexile* strains. Their genomes encode various pathways of central carbohydrate metabolism, including the Embden-Meyerhof pathway (except for a key gene encoding hexokinase/glucokinase (EC 2.7.1.1, EC 2.7.1.2)), gluconeogenesis, the tricarboxylic acid cycle, the non-oxidative pentose phosphate pathway, 5-phospho-α-D-ribose-1-diphosphate (PRPP) biosynthesis, pyruvate oxidation, and sulfur metabolism. Moreover, the novel strains KMM 9835^T^ and *M. soesokkakense* KCTC 32427^T^ were found to contain genes encoding for the archaeal pentose phosphate pathway (M00580). The *Mariniflexile* genomes have the full pathways for the biosynthesis of amino acids (Pro, Lys, Thr, Ile, Ser, Trp, Leu, and His). In all genomes, the synthesis of ornithine lacks a gene for amino-acid N-acetyltransferase (EC 2.3.1.1), which is responsible for glutamate to N-acetylglutamate conversion. However, ornithine could be synthesized through the proline metabolism pathway (M00972). In addition, a gene encoding acetylornithine deacetylase (EC 3.5.1.16), which catalyzes N-acetylornithine to L-ornithine, was absent in the *M. soesokkakense* KCTC 32427^T^ genome. The KMM 9835^T^ genome contains complete degradation pathways, converting methionine to homocysteine, leucine to acetoacetate and acetyl-CoA, proline to glutamate, and a glycine cleavage system. In the pathway of tryptophan metabolism, aminomuconate-semialdehyde dehydrogenase (EC 1.2.1.32) was absent. The hydroxyproline degradation pathway lacked D-hydroxyproline dehydrogenase (EC 1.5.99.-). The pathway for histidine degradation to glutamate was not detected except for formiminoglutamase (EC 3.5.3.8). The pathways for the following cofactor and vitamin synthesis were predicted in KMM 9835^T^: NAD (aspartate and tryptophan-derived), coenzyme A, heme, riboflavin, tetrahydrofolate, and lipoic acid. The menaquinone biosynthetic pathway was short of a succinyl-6-hydroxy-2,4-cyclohexadiene-1-carboxylate synthase (EC 4.2.99.20). The pantothenate biosynthesis pathway was incomplete due to the absence of a 2-dehydropantoate 2-reductase (EC 1.1.1.169). Siroheme is not produced since there is no sirohydrochlorin ferrochelatase (EC 4.99.1.4). The molibdenum cofactor biosynthesis pathway lacks molybdopterin adenylyltransferase (EC 2.7.7.75). Pathways for biotin synthesis were absent in KMM 9835^T^.

### 3.3. CAZymes Repertoires and Predicted PULs Analysis

The dbCAN annotation analysis revealed that strains within the *Mariniflexile* genus possess a highly diverse repertoire of CAZymes and PULs, which may allow them to utilize a wide range of polysaccharides. The proportion of CAZyme-encoding genes in their genomes varied from 5.3% to 7.9%, with the maximum amount predicted in strains TRM1-10 and KMM 9835^T^ (Figure 5a). In the KMM 9835^T^ genome, CAZyme-encoding genes reached 7.3%, comprising 164 glycoside hydrolases (GHs) classified into 52 families, 65 glycosyltransferases (GTs) into 10 families, 29 polysaccharide lyases (PLs) into 9 families, 19 carbohydrate esterases (CEs) into 9 families, and 2 auxiliary activities (AAs) into 2 families (Figure 5a).

The highest number of GHs found in KMM 9835^T^ was related to the GH43 family (19 predicted encoding genes) containing arabinases and xylosidases (Figure 5b). That indicates the potential of the novel strain KMM 9835^T^ to cleave arabinose moieties from xylans and pectins [60]. The greatest number of GH43-encoding genes among those of *Mariniflexile* spp. were also predicted in *Mariniflexile* sp. TRM1-10, isolated from the rhizosphere of tomato (Figure 5b). It can be assumed that strain TRM1-10 is able to utilize pectin, which is a polysaccharide primarily characteristic of land plants. The other abundant GHs of strain KMM 9835^T^ were found to belong to GH2 beta-galactosydases (16 genes) catalyzing the degradation of different oligosaccharides. Members of the GH28 and GH92 families were also widely distributed within the KMM 9835^T^ genome. The GH28 family was represented by 11 putative polygalacturonases that may cleave the backbone glycosidic linkages of pectin using a hydrolytic reaction [61]. The GH92 family contains α-mannosidases responsible for N-glycan cleavage that are common in many flavobacterial species [62]. All of these GHs were found within PULs (Figure 3), with the majority of GH92 members concentrated in PUL3, which is predicted to hydrolyze mucin-rich substrates. Most annotated polysaccharide lyases were classified as PL1 and PL10 families, possessing pectin and pectate lytic activities. Among carbohydrate esterases, the CE8 and CE12 families recognized for facilitating the degradation of pectin by deacetylation and demethylation [63] were the most abundant. The *Mariniflexile* sp. TRM1-10 and *M. soesokkakense* KCTC 32427^T^ genomes shared similarities with the KMM 9835^T^ distribution of CAZyme gene families (GH43, GH28, PL1, PL10, CE8, and CE12), which are responsible for pectin degradation. This may represent part of their adaptation strategy for facilitating microbial consumption of plant polymeric substrates in aquatic environments near shorelines and freshwater sources.

The second most frequent enzyme family in the KMM 9835^T^ CAZome is GTs (65 encoding genes) (Figure 5a). GT2 and GT4 accounted for the highest proportion in the GT families, followed by GT51 in KMM 9835^T^ and the related strain genomes (Figure 5b). GT2 and GT4 have been shown to perform the synthesis of α- and β-glycans and glycoconjugates [64], while GT51 enzymes, known as peptidoglycan glycosyltransferase, take part in the synthesis of murein in both Gram-positive and Gram-negative bacteria [65].

Signal peptide prediction in the KMM 9835^T^ CAZome provided by dbCAN [48] revealed that about half of the total number of GHs, PLs, and CEs genes have signal peptide sequences targeting their products to the periplasmic space. In addition, some secretory CAZymes were predicted to contain CTDs and can be transported by the unique *Bacteroidota* T9SS [11,66], supporting their extracellular role in polysaccharide metabolism.

Based on biochemical characteristics (Table 2), *Mariniflexile* spp., except for *M. gromovii* KMM 6038^T^, showed the ability to hydrolyze starch [1,4,8,10]. Deep genomic analysis revealed that the genomes of novel bacteria KMM 9835^T^, *M. soesokkakense* KCTC 32427^T^, *M. maritimum* KCTC 72895^T^, and *M. fucanivorans* DSM 18792^T^ contain genes encoding for the starch utilization system (Sus) [67], which explains the difference in the substrate utilization abilities of all the above-mentioned bacteria. However, these genes were not found in the *M. gromovii* KMM 6038^T^ genome, which explains the difference in their substrate utilization abilities. In the KMM 9835^T^ genome, the *sus* locus includes six genes, encoding outer membrane SusCD (E/F)G and periplasmic SusAB proteins. SusG is an α-amylase with CBM48, GH13_10, and C-terminal T9SS domains, which allows for hydrolysis of α-1-4 glucosidic linkages at the cell surface. Interestingly, SusC, SusD, hybrid SusE/F, and two neighboring hypothetical proteins shared 75–90% similarity with those from *Tenacibaculum adriaticum* DSM 18961^T^, *Gelidibacter salicanalis* PAMC21136^T^, and *Algibacter agarivorans* JCM 18285^T^. Moreover, this region in the KMM 9835^T^ genome was detected as a HGT region by the AlienHunter tool in Proksee [42]. Neopullulanase SusA (GH13_46) and α-glucosidase SusB (GH97) may be responsible for the degradation of oligosaccharides to maltose. Thus, a full starch utilization system was found in all the *Mariniflexile* genomes except for *M. gromovii* KMM 6038T, which is consistent with the data from the biochemical tests.

Genomic data also confirmed the obtained biochemical characteristics (Table 2) that indicate that *Mariniflexile* spp., with the exception of *M. maritimum* KCTC 72895^T^, is not capable of agar hydrolyzing [1,4,8,10]. The gene encoding beta-agarase (EC 3.2.1.81) that shares 61% similarity to beta-agarase A (GH16, CAZ98338.1) of *Zobellia galactanivorans* Dsij^T^ [68] was found only in the *M. maritimum* KCTC 72895^T^ genome.

Finally, we hypothesize that the environment can influence the gene composition of *Mariniflexile* species, providing them with an adaptive potential to degrade natural polysaccharides specific to the particular ecological niche. The most obvious difference between *Mariniflexile* strains isolated from the seashore or plants (KMM 9835^T^, *Mariniflexile* sp. TRM1-10, and *M. soesokkakense* KCTC 32427^T^) and from sea water or marine organisms (*M. fucanivorans* DSM 18792^T^, *M. gromovii* KCTC 12570^T^, and *Mariniflexile* sp. AS56) is that they have similar CAZomes, despite their distant positions on the phylogenomic tree (Appendix A); KMM 9835^T^ is phylogenetically close to strain AS56 and most distant from the strains TRM1-10 and *M. soesokkakense* KCTC 32427^T^ (Figure 5b).

### 3.4. Morphological, Physiological, and Biochemical Characteristics

Strain KMM 9835^T^ was found to be Gram-negative, aerobic, non-motile bacteria. Colonies were yellow-pigmented shiny ones with regular edges of 2−3 mm in diameter on MA 2216. Electron microscopy observation revealed rod-shaped cells, 1.5−2.0 μm long and 0.7−0.9 μm in diameter, and extracellular material production was observed (Figure 6).

Phenotypic characteristics of strain KMM 9862^T^ are given in Table 2 and Table 3, Appendix A, and in the species description. The bacterium KMM 9835^T^ was able to grow in the narrow salinity range of 0–5% NaCl and at a temperature of 5–36 °C but was not able to utilize carbohydrates sources in the API 20E and API 20NE tests. It should be noted that the novel strain grew slowly without NaCl addition (0% NaCl) and weakly with 5% NaCl. The genomic DNA G+C content of strain KMM 9835^T^ was 32.5 mol% (Table 2).

The dominant menaquinone was MK-6, and the major fatty acids were iso-C15:0, iso-C15:1 ω10c, and C15:0, followed by iso-C17:0 3-OH (Table 3). Fatty acid profiles were similar, with large proportions of iso-C15:0 and C15:0 found in all strains tested (Table 3). Strain KMM 9835^T^ contained a slightly lower amount of anteiso-C15:0 2-OH, and *M. maritimum* KCTC 72895^T^ contained a certain amount of iso-C14:0, iso-C16:1ω6c, iso-C16:0, and iso-C16:0 3-OH compared with other strains tested (Table 3).

The polar lipids of strain KMM 9835^T^ consisted of phosphatidylethanolamine (PE), two unidentified aminolipids (AL1, AL2), an unidentified phospholipid (PL), and six unidentified lipids (L1-L6) (Appendix A). Strain KMM 9835^T^ was close in its polar lipid profile to that of *M. soesokkakense* KCTC 32427^T^, except for the presence of PL, which was not found in the other two strains. The polar lipids of all strains tested included PE, AL1, AL2, and unidentified lipids L1-L6 or L1-L4, as in the case of *M. maritimum* KCTC 72895^T^. The latter additionally contained phosphatidylcholine, which was not present in other strains tested (Appendix A). The presence of iso-C15:0, iso-C15:1, C15:0, iso-C17:0 3-OH, and MK-6, and major polar lipid components, are corroborated with those previously described for *Mariniflexile* species [2,3,4,10].

## 4. Conclusions

The phylogenetic relationships observed on the basis of 16S rRNA gene and whole genome sequences and genetic distinctness as revealed by ANI and dDDH analyses were supported by phenotypic differences of the novel isolate KMM 9835^T^ in its growth temperature and salinity ranges, enzyme activity, and substrate hydrolysis. Differential phenotypic characteristics are indicated in Table 2. Based on the combined phylogenetic evidence and phenotypic characteristics, it is proposed to classify marine sediment strain KMM 9835^T^ as a novel species, *Mariniflexile litorale* sp. nov.

Description of *Mariniflexile litorale* sp. nov.

*Mariniflexile litorale* (li.to.ra’le. L. neut. adj. *litorale*, of the seashore, a shallow-seawater dweller).

Gram-negative, aerobic, oxidase-positive (weak reaction), catalase-positive, rod-shaped non-motile cells, 0.7–0.9 μm in width and 1.5–2.0 μm in length. Grows on MA 2216 or in MB 2216. On MA 2216, it produces hemi-transparent yellow-pigmented shiny smooth colonies with regular edges of 2–3 mm. Does not require NaCl for growth; growth occurs in 0–5 (*w*/*v*) NaCl with an optimum of 2% NaCl; growth in 0% and 5% NaCl is observed as slow and weak, respectively. The temperature range for growth is 5–36 °C, with an optimum of 25–28 °C. Does not grow at 4 °C and 37 °C. The pH range for growth is 5.5–9.5 (optimal pH 6.5–7.5). Positive for hydrolysis of starch, Tweens 20 and 40, and negative for hydrolysis of DNA, gelatin, casein, Tween 80, tyrosine, chitin, and agar, and production of H_2_S from thiosulfate. In the API 20E, negative for the ONPG test, arginine dihydrolase, lysine decarboxylase, ornithine decarboxylase, citrate utilization, H_2_S and urease production under anaerobic conditions, tryptophane deaminase, indole production, acetoin production (Voges-Proskauer reaction), gelatin hydrolysis, and oxidation/fermentation of D-sucrose, D-glucose, D-mannitol, inositol, D-sorbitol, L-rhamnose, D-melibiose, amygdalin, and L-arabinose. According to the API 20NE, positive for the PNPG test and esculin hydrolysis and negative for nitrate reduction, gelatin hydrolysis, indole production, glucose fermentation, arginine dihydrolase, urease, assimilation of D-glucose, D-mannitol, maltose, D-gluconate, L-malate assimilation of D-mannose, L-arabinose, N-acetylglucosamine, caprate, adipate, citrate, and phenylacetate.

Positive API ZYM test results are obtained for alkaline phosphatase, esterase (C4), esterase lipase (C8), leucine arylamidase, valine arylamidase, cystine arylamidase (weak reaction), acid phosphatase, naphthol-AS-BI-phosphohydrolase, β-galactosidase, and N-acetyl-β-glucosaminidase, and negative for lipase (C14), trypsin, α-chymotrypsin, α-galactosidase, α-glucosidase, β-glucosidase, β-glucuronidase, α-mannosidase, and α-fucosidase.

The dominant menaquinone is MK-6, and the major fatty acids are iso-C15:0, iso-C15:1 ω10c, and C15:0. The polar lipids comprise phosphatidylethanolamine, two unidentified aminolipids, an unidentified phospholipid, and six unidentified lipids. The DNA GC content of 32.5% is calculated from the genome sequence.

The DDBJ/GenBank accession numbers for the 16S rRNA gene and genome sequences of strain KMM 9835^T^ are OQ300347 and JASCRQ010000000 (GCF_031128465.2), respectively.

The type strain of the species is strain KMM 9835^T^ (=KCTC 92792^T^), isolated from the sediment sample collected from the Amur Bay of the Sea of Japan seashore, Russia.

## Figures and Tables

**Figure 1 microorganisms-12-01413-f001:**
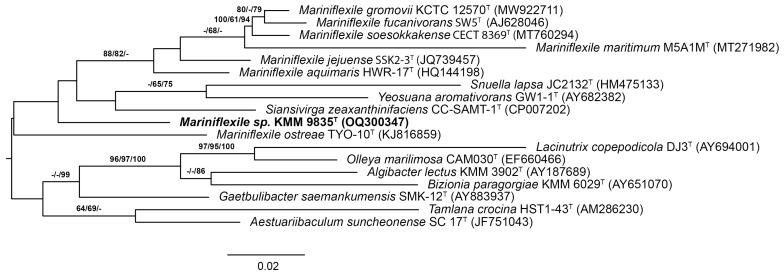
NJ/ML/MP tree based on 16S rRNA gene sequences available from the GenBank database showing relationships between the novel strain KMM 9835^T^ (in bold), *Mariniflexile* species, and related taxa of the family *Flavobacteriaceae*. The NJ tree was reconstructed using the Kimura two-parameter model. The ML tree was inferred under the GTR + GAMMA model. The branches are scaled in terms of the expected number of substitutions per site. The numbers above the branches represent bootstrap values with 1000 replicates larger than 60% (NJ/ML/MP). The bar indicates 0.02 accumulated substitutions per nucleotide position.

**Figure 2 microorganisms-12-01413-f002:**
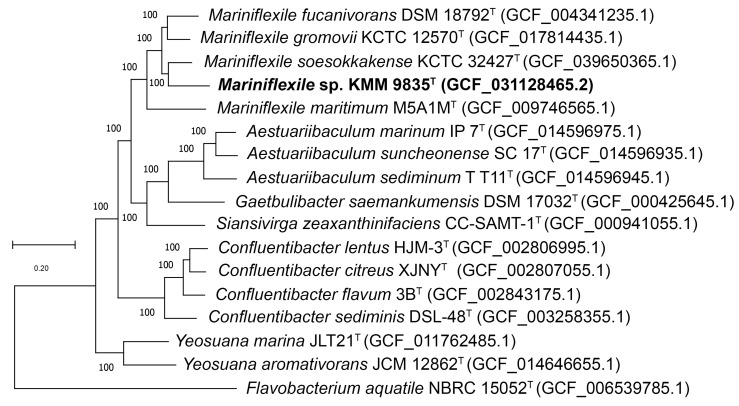
ML tree based on concatenated sequences of 341 translated proteins showing the phylogenetic position of strain KMM 9835^T^ among *Mariniflexile* species and related taxa. The tree was inferred under the PROTCATLG evolutionary model using 100 replicates for bootstrapping. Bar: 0.20 substitutions per amino acid position.

**Figure 3 microorganisms-12-01413-f003:**
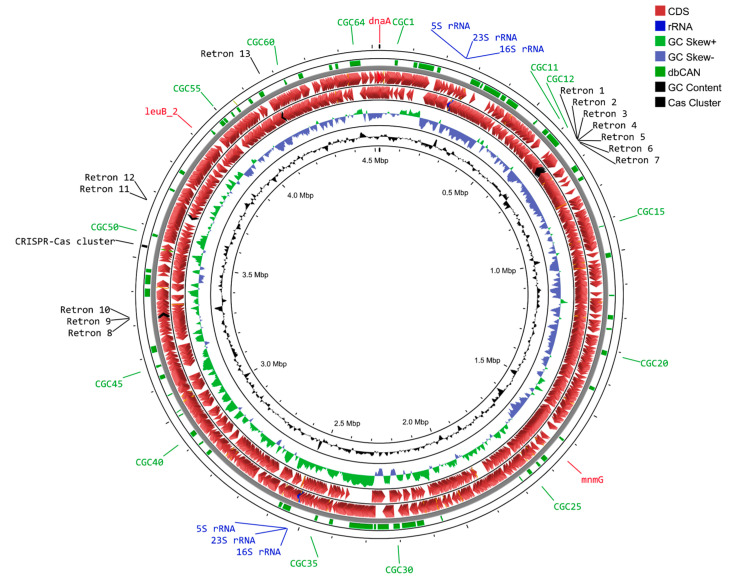
Chromosome map of strain KMM 9835^T^ created using the Proksee server [42]. The scale is shown in megabases (Mbp) on the inside circle. Starting with the inner rings, the first two circles represent GC content (in black) and GC skew (G−C)/(G+C) (in violet blue and light green). The next two dark red circles show reverse and forward strand CDSs. Moving outward, the dark green circle shows PULs designated as CGCs, annotated by the dbCAN server [49]. The outermost circle shows the CRISPR-Cas region (in black). The figure also shows retron-type RNA-directed DNA polymerase (EC 2.7.7.49) (designated as retron 1–13 with black labels), *rrn* operons (blue labels), *oriC* (*leuB*_2 and *dnaA*), and *ter* (*mnmG*) (red labels).

**Figure 4 microorganisms-12-01413-f004:**
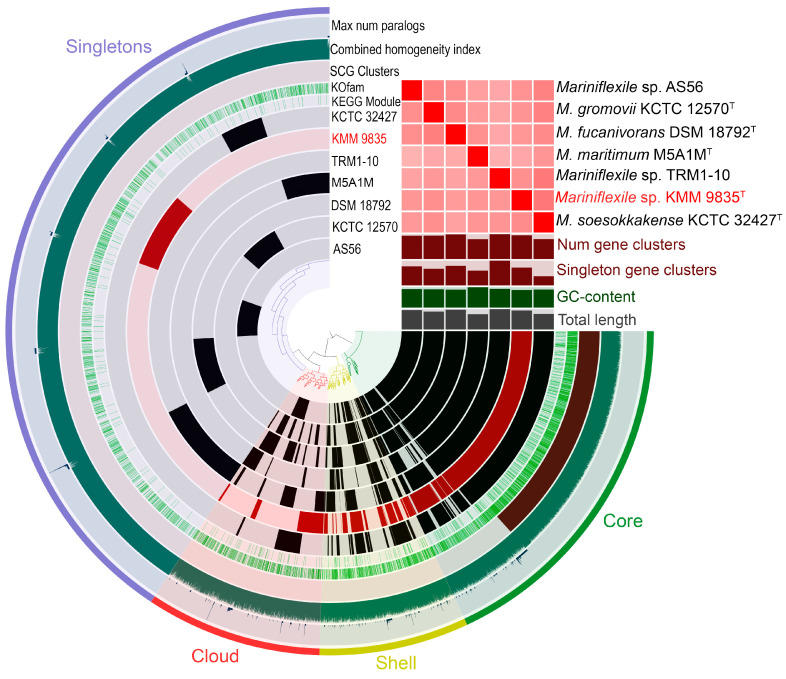
The pan-genome of seven strains of *Mariniflexile* spp. generated with anvi’o [50]. Circle bars represent the presence/absence of 9163 pan-genomic clusters in each genome. Gene clusters are organized as core (green), shell (yellow), cloud (red), and singleton (purple) gene clusters using Euclidian distance and Ward ordination. The heatmap in the upper right corner shows pairwise values of average nucleotide identity (ANI) in percentages. The bars under the heatmap show, relative to each genome, the number of gene clusters (0–3881), number of singleton gene clusters (0–1010), GC-content (0–0.37778), and total length (0–4,858,325). The strain KMM 9835^T^ is colored red. Other information included in the figure comprises the maximum number of paralogs, combined homogeneity index, single-copy gene clusters (SCG clusters), and KOfam and KEGG modules (green and light green circles).

**Figure 5 microorganisms-12-01413-f005:**
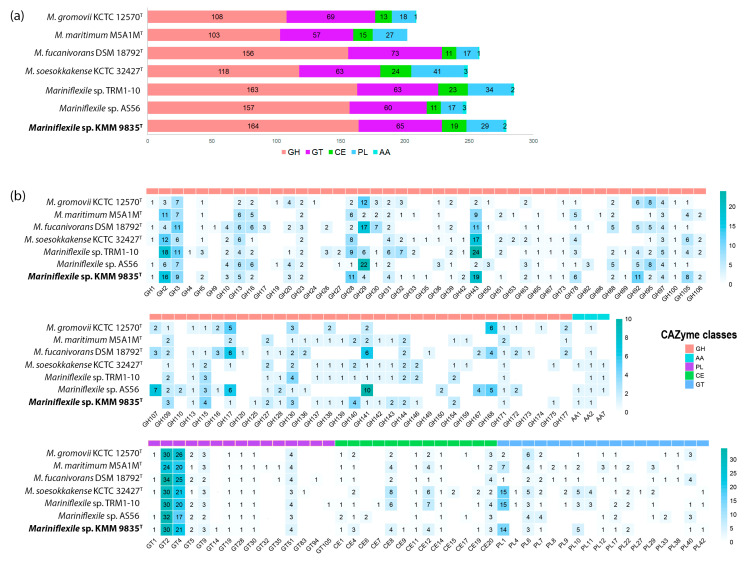
Distribution of CAZymes within the *Mariniflexile* genus. (**a**) Number of CAZyme classes in KMM 9835^T^ and other *Mariniflexile* species. (**b**) Heatmap of CAZyme family abundance in *Mariniflexile* species. GH—glycoside hydrolase, GT—glycosyltransferase, CE—carbohydrate esterase, PL—polysaccharide lyase, AA—auxiliary activity.

**Figure 6 microorganisms-12-01413-f006:**
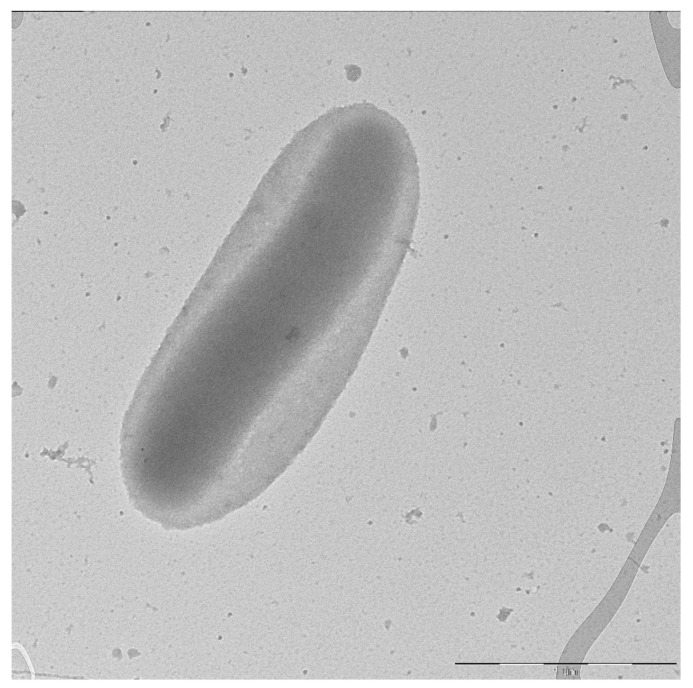
A transmission electron micrograph of strain KMM 9835^T^, grown on MA 2216. Bar, 1 µm.

**Table 1 microorganisms-12-01413-t001:** Genomic features of strains KMM 9835^T^, KCTC 32427^T^, and other *Mariniflexile* strains.

Feature	1	2	3	4	5	6	7
Assembly level	chromosome	contig	scaffold	scaffold	scaffold	chromosome	contig
Genome size (Mb)	4.5	3.7	4.7	4.3	3.7	4.9	4.7
Number of contigs	1	27	29	52	38	1	50
G+C Content (mol%)	32.5	33.5	33	33.5	37.5	34.5	34
N50 (Kb)	4521.4	648.1	432.9	177.4	292.6	4858.3	288.3
L50	1	3	4	7	4	1	6
Coverage	180.0×	63.0×	317.0×	100.0×	100.0×	219.1×	207.0×
Total genes	3839	3158	4104	3770	3157	3987	3789
Protein coding genes	3771	3100	3992	3690	3088	3915	3722
rRNAs (5S/16S/23S)	2/2/2	1/1/1	1/1/1	1/1/1	1/1/1	2/2/2	1/1/1
tRNA	39	36	40	37	38	41	37
checkM completeness (%)	100	100	99.35	98.22	100	100	100
checkM contamination (%)	0.81	0.32	1.05	0.49	1.29	0.32	0.97
WGS project/RefSeq	GCF_031128465.2	JAZHYP01	SLUP01	JAGJCB01	WRPN01	NZ_CP022985	JAUPED01
Genome assembly	ASM3112846v2	ASM3965036v1	ASM434123v1	ASM1781443v1	ASM974656v1	ASM342598v1	ASM3055247v1

Strains: **1**, KMM 9835^T^; **2**, *M. soesokkakense* KCTC 32427^T^; **3**, *M. fucanivorans* DSM 18792^T^; **4**, *M. gromovii* KCTC 12570^T^; **5**, *M. maritimum* M5A1M^T^; **6**, *Mariniflexile* sp. TRM1-10; **7**, *Mariniflexile* sp. AS56.

**Table 2 microorganisms-12-01413-t002:** Differential characteristics of strain 9835^T^ and the type strains of the most closely related *Mariniflexile* species.

Characteristic	1	2	3	4	5
DNA GC content (%) *	32.5	33.5	37.5	33.5	33.0
Growth in:					
5% NaCl	(+)	−	(+)	+	+
6% NaCl	−	−	−	+	−
Growth at:					
5 °C	+	−	−	+	+
36 °C	(+)	(+)	+	+	−
37 °C	−	−	+	+	−
Oxidase reaction	(+)	+	+	−	−
Hydrolysis of:					
Starch	+	+	+	−	+
Agar	−	−	+	−	−
Gelatin	−	−	−	+	−
DNA	−	−	+	(+)	+
Tween-40	+	−	−	−	−
API ZYM tests:					
Esterase C4	+	(+)	+	+	+
Lipase C 14	−	−	−	−	(+)
Valine arylamidase	+	−	+	+	+
Cystine arylamidase	(+)	−	(+)	+	+
Trypsin	−	−	−	+	+
α-chymotrypsin	−	−	−	+	+
α-galactosidase	−	+	+	−	−
α-glucosidase	−	+	+	−	+
β-glucosidase	−	−	+	−	+
α-fucosidase	−	−	−	−	+

Strains: **1**, KMM 9835^T^; **2**, *M. soesokkakense* KCTC 32427^T^ (data were obtained from the present study unless otherwise indicated); **3**, *M. maritimum* KCTC 72895^T^ (data were obtained from the present study unless otherwise indicated); **4**, *M. gromovii* KMM 6038^T^ (data from Nedashkovskaya et al., 2006 [1]; Barbeyron et al., 2008 [10]); **5**, *M. fucanivorans* DSM 18792^T^ (data from Barbeyron et al., 2008 [10]). * The DNA GC contents of the strains KMM 9835^T^, *M. maritimum* KCTC 72895^T^, *M. gromovii* KCTC 12570^T^, and *M. fucanivorans* DSM 18792^T^ were derived from the GenBank. +, positive; −, negative; (+), weak reaction. All strains were positive for hydrolysis of aesculin, alkaline phosphatase, esterase lipase C8, leucine arylamidase, acid phosphatase, and β-galactosidase activities, and negative for production of flexirubin pigments, α-mannosidase, and β-glucuronidase.

**Table 3 microorganisms-12-01413-t003:** Cellular fatty acid composition (%) of strain KMM 9835^T^ and type strains of related *Mariniflexile* species.

Fatty Acid	1	2	3
iso-C_14:0_	0.54	3.26	0.73
iso-C_15:1_ *ω*10*c*	12.74	3.97	9.80
iso-C_15:0_	12.87	15.51	16.96
anteiso-C_15:0_	5.31	6.34	5.71
C_15:1_ *ω*11c	2.13	1.42	1.91
C_15:1_ *ω*6*c*	4.20	3.03	5.25
C_15:0_	11.40	7.59	12.54
iso-C_16:1_ *ω*6*c*	0.37	4.35	0.47
iso-C_16:0_	0.58	4.05	0.62
C_16:1_ *ω*7*c*	4.42	3.71	2.82
iso-C_15:0_ 2-OH	3.26	4.70	1.81
anteiso-C_15:0_ 2-OH	6.02	1.77	1.81
C_16:0_	0.88	1.58	0.81
iso-C_15:0_ 3-OH	6.44	4.67	10.31
anteiso-C_15:0_ 3-OH	1.58	2.59	1.72
iso-C_17:1_ *ω*7*c*	1.27	1.17	0.73
C_15:0_ 3-OH	2.86	1.42	2.73
C_17:1_ *ω*6*c*	1.10	1.52	1.49
C_16:0_ 2-OH	0.07	1.58	0.11
iso-C_16:0_ 3-OH	3.36	11.45	4.42
anteiso-C_16:0_ 3-OH	1.06	1.53	1.20
iso-C_17:0_ 3-OH	9.85	5.50	9.50
anteiso-C_17:0_ 3-OH	1.28	1.11	1.15

Strains: 1, KMM 9835^T^; 2, *M. maritimum* KCTC 72895^T^; 3, *M. soesokkakense* KCTC 32427^T^ (data were obtained from the present study).

## Data Availability

The type strain of the species is strain KMM 9835^T^ (=KCTC 92792^T^), isolated from the sediment sample collected from the Amur Bay of the Sea of Japan seashore, Russia. The DDBJ/GenBank accession numbers for the 16S rRNA gene and genome sequences of strain KMM 9835^T^ are OQ300347 and JASCRQ010000000, respectively. The DDBJ/GenBank accession number for the genome sequence of strain *Mariniflexile soesokkakense* KCTC 32427^T^ is JAZHYP000000000.

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
