# Peer review of "Description and Whole-Genome Sequencing of Mariniflexile litorale sp. nov., Isolated from the Shallow Sediments of the Sea of Japan"

_microorganisms, 2024, doi:10.3390/microorganisms12071413_

Round 1

Reviewer 1 Report

Comments and Suggestions for Authors

The manuscript "Description and whole-genome sequencing of Mariniflexile litorale sp. nov., isolated from the shallow sediments of the Sea of Japan" presents a well-studied with thorough phylogenetic, phenotypic, and genomic analyses that convincingly support the classification of Mariniflexile litorale as a novel species.

The methodology is robust and detailed, the results are clearly presented, and the discussion effectively contextualizes the findings within the field of bacterial taxonomy.  

Therefore, I recommend this manuscript for publication.

Author Response

Thank you very much for reviewing our manuscript. We appreciate your time and your decision.

Reviewer 2 Report

Comments and Suggestions for Authors

Review for Microorganisms 

Overall, the goal of this study is to genetically and physiologically characterize a novel species, Mariniflexile litorale. The authors accomplished this throughout the study. Their methods were appropriate, and they were very thorough in their description of the microbial genome and physiology. The conclusions are appropriate for the scope of the study. However, the paper overall would benefit from including additional context regarding the ecological consequences of some of the genomic and physiological findings. Specifically, how do these genomic and physiological characteristics translate to the observed or predicted ecology of M. litorale, and how is it differentiated from other Mariniflexile spp.?  

Introduction 

Line 48: ‘including from the sea urchin’ 

Line 49-51: This sentence is difficult to interpret as currently written. Consider rewording to emphasize where Mariniflexile fucanivorans has previously been isolated and its ability to degrade sulfated fucans.   

Line 54-55:The second adaptive system utilized uniquely by Bacteroidota 

Line 55: It would be helpful to define Polysaccharide Utilization Loci more explicitly. What are PULs? What ecological benefits might they confer to different species within Bacteroidota? In fact, it might actually be more helpful to introduce this concept, then give the specific example of M. fucanivorans degrading brown algae fucans. 

Line 88: As described earlier 

Materials & Methods 

Line 78: Italicize to be consistent with other subheadings. 

Line 102: Which standards were used in the identifications of FAMEs? 

Line 115-117: Which primers were used to PCR amplify the 16S rRNA gene? What were the reaction conditions? 

Section 2.5: It is unclear which bioinformatic tools were used for which set of sequencing reads, and how each set of sequencing reads is being used in the analysis. More context is needed to understand why and how the sequences are being generated and analyzed. 

Line 129-132 Why were the species sequenced on two different sequencing machines? Why use both short and long reads? 

Line 133: Please provide the parameters that were used with Trimmomatic.  

Line 136: Please provide any parameters that were used in running SPAdes. 

Line 140-141: What parameters/flags were used with samtools to estimate sequencing depth? 

Line 148: What specific parameters were used with the PhyloPhlAn software? 

Line 150-152: Which tool was used for gene prediction? Which databases were used to annotate the genes? Were there any specific parameters used by these tools to annotate the genes? 

Results 

Figure 1: Is this based on the full-length 16S gene? It appears that Mariniflexile sp. KMM 9835 groups with other bacteria that do not belong to the Mariniflexile genus, such as Siansivirga, Yeosuana, and Snuella. There needs to be clear rationale for why this proposed novel species is being placed within the Mariniflexile genus if other genera are included within the tree. Is the idea that this is likely because the 16S rRNA tree provides less confident placement than the protein tree (Figure 2)? If so, could you clarify that? 

Figure 2: How was this tree generated? Please be more explicit with the methods used to generate this figure. Specifically, which 400 proteins were used in figure generation? 

Figure 4: Consider including the numbers used to calculate the bars somewhere in the figure. The bars themselves are difficult to read. 

Line 340-342: Why is this not surprising? Please provide further ecological context for why the presence of these genes is important, or why they’re differentially abundant in different species of Mariniflexile 

Line 362-364: This is the start of a good idea, but the sentence is incomplete. Perhaps expand on the ecological significance of these GTs? 

Line 375-377: This is interesting! Please elaborate on the different Mariniflexile species’ substrate utilization. 

Line 384-385: One more sentence tying together and contextualizing the findings of the different starch utilization genes would be helpful here. What is the big takeaway from this paragraph? 

Lines 417-422: What are these fatty acids for? What is the ecological significance of these findings? It would be helpful to include some additional context for this.

Comments on the Quality of English Language

There were only a few minor areas where I noticed some strange English. It is likely that these errors will be cleared up following my minor edits and with copy editing. 

Author Response

Responses for Reviewer #2: Thank you very much for reviewing our manuscript and for your valuable comments and suggestions.

Comment: Overall, the goal of this study is to genetically and physiologically characterize a novel species, Mariniflexile litorale. The authors accomplished this throughout the study. Their methods were appropriate, and they were very thorough in their description of the microbial genome and physiology. The conclusions are appropriate for the scope of the study. However, the paper overall would benefit from including additional context regarding the ecological consequences of some of the genomic and physiological findings. Specifically, how do these genomic and physiological characteristics translate to the observed or predicted ecology of M. litorale, and how is it differentiated from other Mariniflexile spp.?

Introduction

Line 48: ‘including from the sea urchin’ 

Response: Thank you, it has been corrected (Line 48).

Comment: Line 49-51: This sentence is difficult to interpret as currently written. Consider rewording to emphasize where Mariniflexile fucanivorans has previously been isolated and its ability to degrade sulfated fucans.

Response: The sentence has been rewritten. We hope it is now clearer to understand Mariniflexile fucanivorans isolation (Lines 49-52).

Comment: Line 54-55: “The second adaptive system utilized uniquely by Bacteroidota…” 

Response: Thank you, it has been accepted (Line 55).

Comment: Line 55: It would be helpful to define Polysaccharide Utilization Loci more explicitly. What are PULs? What ecological benefits might they confer to different species within Bacteroidota? In fact, it might actually be more helpful to introduce this concept, then give the specific example of M. fucanivorans degrading brown algae fucans. 

Response: We have added some description of canonic PULs (Lines 57-59), although the definition PULs is described in details in the cited paper #12. Unfortunately, there are no data on studying PUL in this bacterium, as well as in other Mariniflexile spp. We plan to devote the next paper to a comparative genomics analysis of the genus Mariniflexile, taking into account genetic and environmental aspects.

Comment: Line 88: ‘As described earlier’ 

Response: Thank you, it has been accepted (Line 94).

Materials & Methods 

Comment: Line 78: Italicize to be consistent with other subheadings. 

Response: It has been corrected.

Comment: Line 102: Which standards were used in the identifications of FAMEs? 

Response: It was added as “Standard bacterial acid methyl ester mix 47080-U, Supelco, Bellefonte, Pennsylvania, USA” (Lines 110-111).

Comment: Line 115-117: Which primers were used to PCR amplify the 16S rRNA gene? What were the reaction conditions? 

Response: Some requested PCR amplification details have been added (Lines 126-130).

Comment: Section 2.5: It is unclear which bioinformatic tools were used for which set of sequencing reads, and how each set of sequencing reads is being used in the analysis. More context is needed to understand why and how the sequences are being generated and analyzed.

Response: The tools and databases used by these servers are specified in the corresponding cited articles. Is it really necessary to specify them again?

Comment: Line 129-132:  Why were the species sequenced on two different sequencing machines? Why use both short and long reads?

Response: Using both short and long reads allows us to get a complete circular chromosome while retaining high quality. Unfortunately, we did not obtain a sufficient number of long reads for the M. soesokkakense RSSK-9T hybrid genome assembly.

Comment: Line 133: Please provide the parameters that were used with Trimmomatic.  

Response: Trimmomatic was used under the PE flag with HEADCROP: 15; LEADING: 30; TRAILING: 30; SLIDINGWINDOW: 4: 25; MINLEN: 100; AVGQUAL: 25 options.

Comment: Line 136: Please provide any parameters that were used in running SPAdes. 

Response: SPAdes was used with default parameters.

Comment: Line 140-141: What parameters/flags were used with samtools to estimate sequencing depth?

Response: First, reads were aligned with minimap2 under –ax sr (for Illumina) or –ax map-ont (for Oxford Nanopore) flags. Then, samtools sort was used to sort the resulting reads. Consequently, samtools coverage was utilized to obtain statistical measurements for short and long reads separately.

Comment: Line 148: What specific parameters were used with the PhyloPhlAn software? 

Response: PhyloPhlAn was used with –diversity medium flag under the default substitution model and configuration file for 400 genes with a bootstrap of 100 replicates (Lines 167-168).

Comment: Line 150-152: Which tool was used for gene prediction? Which databases were used to annotate the genes? Were there any specific parameters used by these tools to annotate the genes?

Response: Baseline parameters from PGAP, RAST and Prokka were used for CDS calling and annotation.

Results

Comment: Figure 1: Is this based on the full-length 16S gene? It appears that Mariniflexile sp. KMM 9835 groups with other bacteria that do not belong to the Mariniflexile genus, such as Siansivirga, Yeosuana, and Snuella. There needs to be clear rationale for why this proposed novel species is being placed within the Mariniflexile genus if other genera are included within the tree. Is the idea that this is likely because the 16S rRNA tree provides less confident placement than the protein tree (Figure 2)? If so, could you clarify that? 

Response: Yes, this tree was based on particularly full-length 16S rRNA gene sequences. Based on the 16S rRNA analysis, the taxonomic position of strain KMM 9835T was unresolved (low branch support, inconsistent clustering). Therefore, phylogenetic analysis based on concatenated sequences of 400 housekeeping protein genes was conducted to identify it. Generally, this type of analysis is better suited for the task, and produces more accurate results (due to a bigger number of conservative genes being used). It has been also reported in [5] (Garcia-Lopez, M.; Meier-Kolthoff, J.P.; Tindall, B.J.; Gronow, S.; Woyke, T.; Kyrpides, N.C.; Hahnke, R.L.; Goker, M. Analysis of 1,000 Type-Strain Genomes Improves Taxonomic Classification of Bacteroidetes. Front. Microbiol. 2019, 10, 2083. doi:10.3389/fmicb.2019.02083). Therefore, we trust the second tree more than the first.

Comment: Figure 2: How was this tree generated? Please be more explicit with the methods used to generate this figure. Specifically, which 400 proteins were used in figure generation? 

Response: The phylogenomic reconstruction was performed using RAxML program based on 341 universal markers selected by using PhyloPhlAn software version 3.0.1 under the PROTCATLG evolutionary model using 100 replicates for bootstrapping (flag -b +-100) [38] (Segata, N., Börnigen, D., Morgan, X. et al. PhyloPhlAn is a new method for improved phylogenetic and taxonomic placement of microbes. Nat Commun 4, 2304 (2013). https://doi.org/10.1038/ncomms3304). This information has been added to the Section Methods (Lines 166-167) and in Figure 2 (Lines 232-235).

Comment: Figure 4: Consider including the numbers used to calculate the bars somewhere in the figure. The bars themselves are difficult to read. 

Response: The numbers used to calculate the bars are added to the figure 4 below its title (Lines 308-310).

Comment: Line 340-342: Why is this not surprising? Please provide further ecological context for why the presence of these genes is important, or why they’re differentially abundant in different species of Mariniflexile.

Response: Pectin is a polysaccharide primarily characteristic for land plants, so the Mariniflexile sp. TRM1-10 isolate, which inhabits the tomato rhizosphere, would have unlimited access to pectin, and probably use it as a substrate due to the presence of enzyme genes of GH43, GH28, PL1, PL10, CE8, and CE12 families (Lines 377-379). Considering your proposal, we have added a sentence as “It can be assumed that strain TRM1-10 is able to utilize pectin, which is a polysaccharide primarily characteristic for land plants.” (Lines 365-366). We suggest that environmental could influence on the gene content of bacteria, providing their adaptive potentials for various natural polysaccharides degradation. The different content of these genes in different Mariniflexile species is obvious, but at the moment we cannot draw any conclusions about why this is so without an in-depth analysis of the metabolic potential of the strains and how it relates to the ecological niche.

Comment: Line 362-364: This is the start of a good idea, but the sentence is incomplete. Perhaps expand on the ecological significance of these GTs? 

Response: No doubt that GTs might have special ecological significance due to their ability to synthesize activated sugars, O-antigenic, exo- and capsular polysaccharide, glycan and so on. It indicates on protective and adaptive functions. However, in that field nothing has been known. We will try to analyze strain GTs gene spectrum depend on their origin isolation in the next paper of a comparative genomics analysis of the genus Mariniflexile.

Comment: Line 375-377: This is interesting! Please elaborate on the different Mariniflexile species’ substrate utilization. 

Response: These data can be seen in the table below (Table 2, Line 442) and in the corresponding cited publications.

Comment: Line 384-385: One more sentence tying together and contextualizing the findings of the different starch utilization genes would be helpful here. What is the big takeaway from this paragraph? 

Response: The big takeaway from this paragraph is that the full starch utilization system was found in all the Mariniflexile genomes except for M. gromovii KMM 6038T, which is consistent with the data from the biochemical tests.

Comment: Lines 417-422: What are these fatty acids for? What is the ecological significance of these findings? It would be helpful to include some additional context for this.

Response: Of course, lipids being a significant component of a bacterial cell wall to be of importance from ecological view point, providing adaptive functions of bacterial cell. But our study devotes to description of a novel bacterial species, where fatty acid content analysis is a routine procedure in the identification and taxonomy of bacteria. It will be a subject of our further paper.

In response to the main question concern to how “these genomic and physiological characteristics translate to the observed or predicted ecology of M. litorale, and how is it differentiated from other Mariniflexile spp.” Finally, we hypothesize that the environment can influence the gene composition of Mariniflexile species, providing them with an adaptive potential to degrade natural polysaccharides specific for the particular ecological niche. The most obvious difference between Mariniflexile strains isolated from the seashore or plants (KMM 9835T, Mariniflexile sp. TRM1-10, M. soesokkakense KCTC 32427T) and from sea water or marine organisms (M. fucanivorans DSM 18792T, M. gromovii KCTC 12570T, Mariniflexile sp. AS56) is that they have similar CAZomes, despite their distant positions on the phylogenomic tree (Figure S2); KMM 9835T is phylogenetically close to strain AS56 and most distant from the strains TRM1-10, M. soesokkakense KCTC 32427T(Figure 5B). (Lines 420-428).

Comment: There were only a few minor areas where I noticed some strange English. It is likely that these errors will be cleared up following my minor edits and with copy editing.

Response: We agree with the errors you found and have corrected them. Thank you for your help.
